# The *Canticle of the Creatures* by Francis of Assisi (1181/82–1226) and the Care of Our Common Home

Isidro Pereira Lamelas 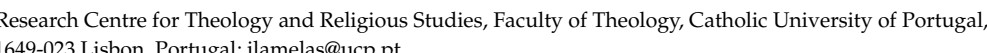

Research Centre for Theology and Religious Studies, Faculty of Theology, Catholic University of Portugal, 1649-023 Lisbon, Portugal; ilamelas@ucp.pt

**Abstract:** In the present essay, we want to show how the *Canticle of the Creatures*, which we might call "*The Canticle of Universal Brotherhood*", is much more than the *Canticle of Brother Sun* or of a single man. The author himself is much more than the exceptional case of a nature-friendly medieval saint who, therefore, continues to inspire the promoters of ecology and, especially after the papal encyclical *Laudato Si'*, constitutes the ecumenical matrix for the care of our common home. To this end, in this paper, we focus on two moments that, in the construction of the tutelary figure of Francis of Assisi, constitute a kind of diptych or portals which open and recapitulate the reconstructive intuition he bequeathed to us: (1) *the vocational moment: Go and repair my house*; and (2) the *testamentary moment*, in which the *Founder*, who never wanted to found anything, legates his manifesto for building the common home as a universal brotherhood, turning the "stones" into a *canticle*. Above all, we want to highlight the relevance of Franciscan spirituality, expressed particularly in the *Canticle of the Creatures*, and thus the Franciscan aesthetics for the modern ecology.

**Keywords:** Francis of Assisi; common home; *Canticle of the Creatures*; ecology; *Laudato Si'*; creatures

## 1. Francis: From Assisi to Rome

The earliest hagiography presents Francis of Assisi as "a new man, one from another world" (*novus certe homo et alterius saeculi videbatur*) (1C 82; Habig 1972, p. 297; see: LM 12.8; TC 54). Eight centuries later, in the encyclical *Laudato Si'* (*LS*), the Bishop of Rome, his namesake, explicitly presents the Saint of Assisi as a "*beautiful and motivating model*" for an integral ecology, beginning with "this gracious hymn that reminds us that our common home can be compared to a sister with whom we share our existence" and "to a good mother" (LS 1), quoting, at the opening of the encyclical, the ninth stanza of the *Canticle of the Creatures* (CC). From the title (*incipit*) *Laudato Si'*, taken from the *Canticle of the Creatures*, the Saint of Assisi appears as a "patron" and model appreciated even by non-Christians: "I believe that Francis is the example par excellence of care for the fragile and for an integral ecology, lived with joy and authenticity" (*LS* 10).

We recall that, in the Apostolic Exhortation *Evangelium Gaudium*, the Pope assumed that "like St. Francis of Assisi, we are called to care for the fragility of the people and of the world in which we live" (216). In no. 87 of *Laudato Si'*, the text of the "gracious canticle of St. Francis of Assisi that inspires the entire papal document" (Nuñez 2016, pp. 91–150) is transcribed almost in its entirety (stanzas 1–8).

This progressive presence[1] of the thought of Francis of Assisi in the pontifical magisterium that culminates in the *Fratelli Tutti* (1–3) undoubtedly marks a new orientation in the Catholic approach to ecological issues (Revol 2021, p. 118). After a long period in which the Franciscan intuition was under the shadow of St. Augustine's and St. Thomas' theological rationalism, we owe to the present Jesuit Pope the merit of reintroducing the Franciscan worldview in the style and in the *forma mentis* of the Church. More than a "novelty", we are facing a rediscovery, as we propose to demonstrate.

Based on early sources of the founding acts of Francis of Assisi, we aim to deepen the reasons why Franciscan rhetoric continues to be the inspiring "difference" for the care

of our common home, especially after the encyclical *Laudato Si'* (*LS*) dedicated to "Sister Earth", without ignoring the "Franciscan question" (historical, literary, and hermeneutical issues relating to the biographies of St Francis written during the first century of the Franciscan movement; from the 19th century onwards, historical and philological critics subjected medieval Franciscan sources to serious scrutiny, leading to a distinction between the historical Francis of Assisi and the Saint of hagiography "constructed" under pressure from the Roman Curia and in accordance with hagiographic clichés. The Calvinist pastor Paul Sabatier stood out as a pioneer of the Franciscan question, with his biography of Francis published in 1894 (Sabatier 1931, p. III)) (Pásztor 2000, pp. 16–21; Uribe 1999, pp. 19–38). In this essay, we focus on two founding moments in the life of the Saint of Assisi which we consider programmatic in shaping Francis's image and his project of the common home in the form of *universal fraternitas* (cf. Le Goff 2007, pp. 105–10). The first moment involves receiving the order, marking the commencement of everything: *Go, Francisco, and repair my house*. The second moment coincides with the culmination of a reconstructed and reconstructive life, recapitulated in the last "stone" in the form of a *Canticle* (*of the Creatures*). In fact, we can see in the *Canticle* of Francis the *culmen* or perfect cap of a "house built on the rock" (Mt 7:24) and the proposal to take care of the common House/Fraternity of universal and perennial value.

We are foremost interested in showing how the canticle *Laudato Si'* condenses a program bequeathed to Western culture that will determine the heuristic lines of an ecological attitude towards nature and both rational and irrational beings. What Francis of Assisi left us was a vision on life that is not only a simple worldview or philosophical system but, rather, a worldview in the full sense of the term, filled with intuitions about Man's place and destiny in the cosmos. There is, in fact, a declared intentionality in Francis's intervention: more than a circumstantial intuition, there is something programmatic in the CC, serving as a paradigmatic guide for future reflection on the place of Man in the world (Gonçalves 1970).

*Francisce, Vade, Repara Domum Meam: Restoring the Ruined House*

The most reliable sources place Francis's famous colloquy with the Crucifix of San Damiano in the autumn of 1205. The *Legend of the three companions* recounts that "a few days after" the "kiss of the leper", while Francis (TC IV, 11) "was walking near the church of San Damiano, an inner voice bade him go in and pray. He obeyed, and kneeling before an image of Crucified Savior, he began to pray most devoutly. A tender, compassionate voice then spoke to him: 'Francis, do you not see that my house is falling into ruin? Go, and repair it for me' (*Francisce, nonne vides quod domus mea destruitur? Vade igitur et repara illam mihi*). Trembling and amazed Francis replied, 'Gladly I will do so, O Lord'" (tremens ac stupens ait: "Libenter faciam, Domine; TC, V, 13; Habig 1972, p. 903). Driven by this readiness and by the context (an abandoned and ruined house–church), he thought it was a matter of "repairing" the material building. He went on for a while, playing the part of a bricklayer, collecting stones in the streets, saying, "shouting" even, "whoever gives me one stone will have one reward; two stones, two rewards; three stones, a treble reward" (TC 21, Habig 1972, p. 911). It is more of an originality of Franciscan begging: instead of asking for bread, he asks for stones. Probably without knowing it, Francis was already acting in a double sense: he was not just building the church of St. Damian, the future hub of his *Fraternitas*, but he was already laying the foundation on the new world of universal fraternity, based on the cosmic man and the cosmos inhabited by a single human and non-human family (Chesterton 2013, p. 74).

The same source adds that "it was his enthusiasm that led him to utter words like these and others" (*ibid.*), concluding with the following significant comment: "It would be difficult to specify all the hard work that had to be done to restore the church; he who at home had been coddled, whereas now he carried a load of stones on his own shoulders" (TC VII, 21; Habig 1972, p. 912).

Thomas of Celano, the first and most reliable biographer (Pásztor 2000, pp. 87–88, 105), uses the same sources but adds some details: feeling impelled to enter a small church "almost ruined and abandoned" (2C, 10–11; TC 13–14; Habig 1972, pp. 370–71; 903–4), Francis kneels and prays. Immediately, he saw the lips of the Crucified One move, and eye to eye, Francis heard the same command: *Francisce, vade, repara domum meam, quae, ut cernis, tota destruitur*.

St. Bonaventure frames this episode after Francis leaves his [father's] home: "he left home to meditate out-of-doors" (LM II, 1; Habig 1972, p. 640). Once again, we have the connection between the essential elements of the reconstruction of Francis's life: leaving the house, meditating amongst and with *the creatures* to re-enter the *house*. This is what happened: he entered the little church of San Damiano, "which was threatening to collapse", where he "heard a voice coming from the crucifix and telling him three times: *Francisce, vade et repara domum meam, quae, ut cernis, tota destruitur!*" (LM II, 1; Habig 1972, pp. 640, 370).

Biographers agree that this is a key moment in Francis's life and the future movement it inspired. We underline the sudden passage from astonishment to action: "he immediately proposes to obey the order received, concentrating all one's forces on its execution" (*ibid.*). It is symptomatic that "the first work that Francis undertook, after he had gained his freedom from the hand of his carnally minded father was to build a house for the God" (1C 18; see 2 C 10; Habig 1972, p. 243).

Francis's behavior has led some to consider him "crazy", not realizing that his gestures are inscribed in the field of a performative symbolism that transcends the narrated facts: "Crazy", he himself assumed, as with all symbols of restorative inversion (Merino 2004, pp. 23–30). What Francis is doing is rebuilding, first the "house" of his own life and person, and consequently the house of the Church, of Society, and of the World that he now carries "on his own back", gathering the stones that some *give* and others *throw*. Indeed, to restore our common home is also to restore it *communally*.

Therefore, we can detect a constructive dynamism since the very first steps of the Franciscan project. These are marked by the following notes: the crisis (*ruin*) as an opportunity and mission; the enthusiasm that leads to concrete action, with a local micro-meaning but also with a universal signifier, which helps to explain that the action precedes the project itself; the repair work of the "house" (*domus*) as a collective and networked effort: *Whoever gives me a stone...*; and, finally, "the reward", which matters and teaches that what moves us is not the fear of the ruin of the *house* but the benefits of a job that impact everyone. The biographer adds that "many workers worked with him" on this construction, responding to the call of the *Poverello* (TC VII, 24; Habig 1972, p. 913). Francis knew of other ways to "build" houses or churches: "have them built", "finance the construction", and do it at the expense of others. That was what the majority did, including the ecclesiastics. He chooses his original mode: the way to build is... building with your own hands, involving everyone.

After about three years dedicated to the restoration of the material patrimony, Francis realized, reading the Gospel of the Mission, that the object to be "restored" was something else: "divinely inspired, he began to speak in public very simply of penitence and the life of evangelical perfection" (TC VIII, 25; Habig 1972, p. 915). It is worth remembering that the word *paenitentia* refers to the Greek *metanoia*, which means a radical change in mentality and habits. This religious language, which is less obvious today, took on a dense political and cultural meaning in Francis's time. When the saint speaks of "doing penance" (T 1; 2R 7:2; Habig 1972, p. 62), he means, in today's terms, "to bring about a radical change of mentality and to operate accordingly". This *change* has been applied on several fronts: in relation to work and money (*paupertas*), in the social sphere (*minors*), in relation to all creatures (*omnes fratres*), regarding the monastic model (*peregrini et edvenae*), etc.

The mission of restoring the common home begins with the conversion (*metanoia*) of those who live in it (*LS* 216–221). This is, in fact, the main "construction" that Francis taught his followers: "As a good and experienced master-builder, he also wished to build his house, that is, his Order, on the firm foundation of holy poverty" (MP 55; Habig 1972,

p. 1178). He had a very special esteem for the "Porziuncola", which he wanted as the cradle and heart of the Order, precisely because it meant a "small portion" (LP 8–9; MP VIII) to remind us that the friars should not build large houses in order to live in them (LP 12), nor should they appropriate land and buildings (MP 55).

It seems that, in the mind of the Saint of Assisi is vivid the warning of the prophets (cf. Haggai 1:4–11). Especially when in Chapter VI of the *Rule*, concerning the "way of working" (*modo laborandi*), he writes: "The friars are to appropriate nothing for themselves, neither a house, nor a place, nor anything else. But as strangers and pilgrims in this world, who serve God in poverty and humility (*Fratres nihil sibi approprient nec domum nec locum nec aliquam rem. Et tanquam peregrini et advenae in hoc saeculo in paupertate et humilitate Domino famulantes* (2R, VI, 1–2; Esser 1978, p. 231; Habig 1972, p. 61). In the *Rule* of 1221, he takes up the subject, emphasizing that "all the friars… they should be the least and subordinate to everyone in the same house" (sed sint minors et submit omnibus, qui in in eadem domo sunt; 1R 7:1; Esser 1978, p. 253; Habig 1972, p. 37). If it is true that these norms "of the house" were written with the *Fraternitas* in mind, they refer to a paradigm which Francis, who never wanted to find an Order, proposes to everyone. In fact, he does not think of or want a house *for himself* or for his brothers but a new way of being and inhabiting the world not based on *possession* or *dominion*. That is why he tore down some houses that his confreres tried to build because they were contrary to poverty (2 C 57; LM 7.2; MP 7).

As such, St. Francis's *Rule*, unlike others, does not dedicate a chapter, or even a few lines, to the building of houses or convents. Rather, it dedicates a special chapter to teach "*how the brethren are to go through the world*" (1R 14: *Quomodo fratres debeant ire per mundum*), asking them to be above all peacemakers: "Whatever house they enter, they should say, 'Peace to this house'" (2R 3; Habig 1972, p. 60). And when contemporaries expressed their astonishment at this alternative way of life, the followers of the *Poverello* recall his words: "Our cloister is the whole world" (SC 63; Habig 1972, p. 1593). A contemporary witness, strange to the Franciscan movement, James of Vitry, bishop (1216) and then cardinal (1228), attentive to the new religious phenomena of his time, speaks of the Franciscan movement as a "cloister the size of the world".[2]

## 2. Aesthetics and Ecology

As Pope Francis reminds us, "the relationship between an appropriate aesthetic education and the preservation of a healthy environment must never be neglected. Paying attention to beauty and loving it helps us to get out of utilitarian pragmatism" (*LS* 215).

It is not only in his *Canticle* that Francis of Assisi proves that aesthetics and ecology go hand in hand (cf. K. Armstrong 2023, p. 130). Fortunately, even today, we can appreciate and contemplate the challenging beauty of the Crucifix or icon that he "ordained" to St. Francis:[3] *Francisce, vade, repara domum meam!* The biographer adds that, "in the meantime, [Francis] did not cease to watch over the holy image of the prodigy, nor to carry out the command he had received" (2C VI, 11). It is no coincidence that this exceptional and inaugural contemplative experience aligns with a radical rupture from the mercantilist paradigm of Peter Bernardone (cf. LM I, 4). "Stripped of all the things of the world, he consecrated himself henceforth to the works of righteousness" (TC VII, 21): from the "pragmatism and utilitarianism" of the paternal *domus*, he passes to the new mission of "restorer" of the house of God which, for him, is the common home. In this context, one can understand the hostilities that he suffered from the "house" of his father, Pedro Bernardone.

The words of St. Bonaventure are equally enlightening to this purpose: "The first step taken" was to "detach oneself from the fortune of one's father's house… he began to organize his life in such a way as to exchange the affairs of the world for the affairs of heaven, in the manner of an evangelical merchant" (LM 4). This new "trade" of evangelical inspiration is well developed in the work *Sacrum commercium Sancti Francisci Domina Paupertate*, written between 1250 and 1260.

José Saramago, in his play *The Second Life of Francis of Assisi*, puts these words in the mouth of the Saint: "Quando, para me entregar à pobreza, renunciei à minha herança,

também renunciei a um pai que nada mais tinha para dar-me que esses bens de vaidade. Para ele, a herança e o filho eram o mesmo. Porque o filho não quis herança, a herança deixou de querer o filho. Ouçam a palavra do dinheiro. . .".[4]

The "inheritance" we are talking about here does not only designate material heritage but a whole society model that, for the first time, began to function around money.[5] On the other hand, the paradigm of the Saint of Assisi commands that "the friars who have a trade should work at it... and that they may receive for their work all that is necessary except money" (1R 7; Habig 1972, p. 37).

If the Christ of San Damiano is the expression of *via pulchritudinis*: *Tu es pulchritude*, *"You are beauty"*, repeats St. Francis when he prays (LD 4. 5; Esser 1978, p. 90; Habig 1972, pp. 125, 126). Then, in the *Canticle of the Creatures*, we can appreciate the masterpiece of a life invested in the development of the spiritual senses that made the ascetic an aesthete, and the contemplative a recreationist. A jubilee and profoundly playful man, Francis uses music and poetry to recover the optimism of those who saw in the Genesis account a symphony in six days (*Hexaemeron*), where every creative act is defined as "beautiful" (*kalón*) [and not just "good"/*agathon*]. For Francis, "Beauty" is one of the names of God, as Pseudo-Dionysius had taught before him (*De ecclesiastica hierarchia*, III, 7.11). In the *Poverello*, the statement of Gregory of Nazianzus is fulfilled, according to which "God made man the singer of his radiance" (Migne n.d., PG 37, p. 1327). Contemplative art is thus at the heart of cosmology, as a way of seeing and listening to the "inner word" of each thing (see Evdokìmov 1990, p. 38). Francis knows the language of animals (1C 166–171), plants (1C 81), and the brothers wind and fire (LP 49; 81; LM V, 9). It was this ability to "listen" that disposed him to hear the voice of God, as well as the voice of all creatures.

By this, we do not mean that our Saint was unaware of the dramas and shadows that inhabit the world and the human heart. But the excessively human nature that led him to embrace the leper and the poor, to embrace the cross of the world's Calvaries all the way to the Alverne, did not prevent him from experiencing the beauty of the Good in the light of the High Good and Trinitarian love. We must not forget that he composed his *Canticle* precisely at the climax of the *via dolorosa* that characterized the last days of his life.

This shows us that the care of the world and of creatures involves contemplative re-enchantment and the exercise of the spiritual senses.

## 2.1. Canticle of the Creatures: A Manifesto for the Care of Our Common Home

The *Song of the Creatures,* or *Praises of Creatures* (2C 96; MP 100; Habig 1972, pp. 533, 1236), or, according to other sources, *Canticle of Brother Sun* (LP 43; MP 120; 123; Habig 1972, pp. 1020–23, 1258–60, 1264) [hereinafter CC], composed in the Umbrian dialect, is the ultimate expression "of the cosmic experience which, in turn, led him to reach the great depths of the sacred" (Merino 2007, p. 19).

The ancient sources (MP 100; 120; 1C 109; 123; 2C 217; LM IX, 1; LP 43, 44, 51, 100) and manuscript tradition leave no doubt as to its author. This hymn to creatures is the fruit of a rich and festive personality, singularly communicative, who lived life intensely (TC 12; 1C 6). However, we are not only dealing with a mystic but also with a creator of environments and currents: "He brought a new springtime into the universe", his companions attest (*ibid.*; cf. 1C 89; Habig 1972, p. 304). In fact, the CC not only mirrors the interior landscape of the *Poverello*; it also reflects his extroverted side that is capable of communicating discursively, poetically, and musically. Francis's inner world is in tune with the outer world because his own psychic and spiritual structure is endowed with a special intimacy and consanguinity with creatures, in a synthesis that is difficult to overcome between interior archaeology and exterior ecology (Merino 1991, p. 150; 2007, pp. 16–17; Nuñez 2016, p. 113).

The essentially positive approach to the created world and all that inhabits it is based on the profoundly positive conception of the Author of Creation: *Tu es bonum, omne bonum, summum bonum;* "You are Good, all Good, supreme Good", Francis prays (LD 3; Esser 90; Habig 1972, p. 125). From this theological basis, we have a profoundly anti-dualist

ontology and anthropology. Man can enter into harmony with all creatures in the goodness that makes them "sisters" and in the fraternity that makes them beautiful.

*Laudato sie, mi' Signore, tucte le tue criature,* thus begins the *Canticle* that *Poverello* composed in the same language used by the simple people of Umbria at the time. He does not "sing" in Latin or in the language of the learned but in the dialect of everyday relationships, wanting to make this poem a popular song. Like its author, this poetry is born "from below", from the earth, in the language of the people, because it wants to be an echo of the voice of all *creatures*. We can affirm, with Friar Fernando Félix Lopes, that the *Canticle of Brother Sol* "was the new sermon that the Saint sent to his friars to preach throughout the world, to fraternize all humanity" (Lopes 1978, p. 464).

The original text still has a particular strength today, not least because, in Old Italian, *criature* is a word with a filial meaning: it designates not only creatures in general but also children. The *Laudato Si'* of this *Canticle* is the filial praise of all *creature-sisters* to God, who, being immensely powerful and sublime, is above all the Goodness that creates and gives life (PLP, 2; PG 3; PBO 11). In all creation, Francis "recognizes the Supreme Beauty, for he heard them all proclaimed: 'He who created us is infinitely good' (2C, 165). The emphasis placed on *all* creatures points to the universal dimension of fraternity proposed by the "universal Brother"[6] (*LS* 221).

Composed and set to music by the Saint of "perfect joy" (LF 8; MP 96; 1R 7; Habig 1972, pp. 38, 1218–320, 1230; Merino 2004, pp. 61–68) as a *canticum* to be effectively sung as a complement to the popular preaching of the friars, these *Laudes criaturarum* were born in the winter of 1225, precisely in the most painful period of Francis's life: the year before his death, sick, almost blind, already marked by the stigmata received in the Alverne in September 1224 (MP 100), removed from the leadership of the Order he had founded, and foreseeing the conflicts that would arise in it after his death. In the dark night of the greatest suffering, Brother Sun, Sister Moon, the stars, and all creatures sing, evoking in each one of them something that is beyond them. According to Francis, "Brother Sun" illuminates us by day, but it is God "who enlightens us through him" (CC 7; Habig 1972, p. 130), even in the middle of the night. That is why the blind poet can see what ordinary eyes cannot.

We also add that, far from being an improvised or momentary piece, the CC, in G major, is the expression of a whole life filled with inspiration. In fact, from a young age, Francis liked to recite and sing the poems of the Provençal rhymers, whose language he knew well (2C 127). Biographers agree in emphasizing Francis's poetic soul (MP 93; 2C 7; 127) and his *troubadour*'s streak (1C 16; MP 82; 1R 21; Habig 1972, pp. 46–47, 242, 1215), composing with ease improvised verses and songs that made him known as the jester, *giullare di Dio* (Merino 2004, pp. 61–64), but have been lost to time. Endowed with a "vibrant and sweet voice, clear and sonorous" (1C 83; Habig 1972, p. 298), he used to sing with his boisterous companions in the streets of Assisi (2C 7). After renouncing his father's inheritance, he went into the woods, "singing the praises of the Creator" (1C 16; Habig 1972, p. 242). His new life of poverty and penance, far from weakening his singing, gave him new motives and inspiration (2C 127). As Eloi Leclerc says, "in Francis the Gospel of poverty is identical to the Gospel of the Canticle. He is the poor who sings" (*Canto*, in *Dizionario Francescano*, 115). When, in the squares of Assisi, he asked for stones to restore the church of San Damiano, he did so by singing (2 C 13); he sings as he makes his way to Rome in the summer of 1209 with his first companions to obtain papal approval of his *rule* of life; together with Friar Egidio, he sets out for the first mission, singing (TC 33). It is not surprising, then, that, even in the final hour, he welcomes death by singing: *mortem cantando suscepit* (1C 117; Habig 1972, p. 331). Even after the CC, he wrote some "words set to music" to "console and edify the Poor Ladies (Poor Clares), knowing how much they suffered because of their illness" (MP 90; Habig 1972, pp. 1223–24): "he spent the few days that remained until his death in praise, inviting his companions whom he loved so much to praise Christ with him... He also invited all creatures to praise God" (2C 217; Habig 1972, p. 536). The *Canticum vel Laudes creaturarum* is thus the culmination of many

other unwritten *canticles* and lauds, which, as Leclerc writes, "accompany, like a refrain, the whole life of Francis of Assisi" (Leclerc 1970, p. 9).

We can say that Francis's originality lies precisely in understanding and living life as a *canticle* (*laus*) to the Creator: "All my brothers and sisters, when they please, can with the blessing of God proclaim among all men this exhortation and laud... to the Creator of all things" (1R 21:1–2; Habig 1972, pp. 46–47). And once again, aesthetics and ecology go hand in hand. "We are", Francis said, "the troubadours of the Lord; we sing his praises... to lift up the heart of man and lead him to spiritual joy/enjoyment" (MP 100; Habig 1972, p. 1236).

We know that the CC is part of the line of the Canticles (Dan 3:51–90) and biblical Psalms of praise (cf. Ps 148; 1C 80), as a poetic liturgy in which man gives voice to all creatures (Sabatelli, 34). Francis, however, expresses in new words the enchanting relationship between the Creator and his works and their relationship as sisters. For the author of the CC, all creatures possess a "significance" from their Creator and are a "mirror" of his goodness and supreme beauty (2C 165; Habig 1972, pp. 494–95). It "revolves around the idea of a new supernatural light on natural things, which means not the rejection but the recovery of natural things" (Chesterton 2013, p. 73). Creatures are "semaphores of transcendence" (Merino 1991, p. 152) with their own dignity and rights. Before being "goods" in the sense of merchandise, they are resonances of the Highest Good. As the current Bishop of Rome acknowledges, "Francis proposes that we recognize nature as a splendid book in which God speaks and transmits something of his beauty and goodness" (*LS* 12). This is because, as Thomas of Celano explains, "the goodness which is at the origin of all things and which one day will be everything in all of them, was already manifested in this life in the eyes of the saint [Francis] clearly total in all of them" (2C 165; Habig 1972, pp. 494–95).

This is not the place to delve into the aesthetic value of the literary piece by the poet–minstrel of Assisi, who, in these verses, takes care of the sonority of his *canticle*, starting with the musicality derived from the insistent anaphora, *Laudato si'... Laudato si'...*, and continued in the rhymes: *Signore/honore, stelle/belle, vento/sostentamento, rengratiate/humilitate*. Note also the numerous assonances, including *sole/splendor, acqua/casta, terra/governa/herba*, and the effect of insistence on some stresses, as in the description of *frate focu... iocondo, robust and strong...* In the adjective *iocondo*, which speaks of service and joy (*iuvare, iucundus*), overflows the *perfect joy* of Franciscans *Fioretti* and the whole *Canticle*. Finally, note the impressionistic touches of the plastic images with which he evokes the variety of fruits (*diversi frutti*) or the coloring of flowers and meadows (*coloriti fiori et herba*; CC 9). The biographer confirms that, when the saint "was enraptured by admiring the beauty of the flowers or inhaling their perfume... and when he found many flowers together, he preached to them and invited them to praise their Lord, as if they were endowed with reason" (1C 81). In these details, we perceive the special sensitivity to cosmic beauty and the contemplative gifts in the face of the *Mother Earth*, who already had a great impact on their contemporaries (1C 81; LM VII, 6).

The *Florinhas* (*Little flowers*), of tenuous historical value, but which we can read as a poetic and symbolic interpretation of the *Franciscan persona* and expressing the "positive and joyful" vision that "overcomes the pessimistic vision typical of the High Middle Ages" (Uribe 1999, p. 427), shows that the sensitive soul of the *Poverello* contemplated the beauties created, without dwelling on them: "in all creatures he sang the Creator... He rejoiced with joy in all the works that came out of the hand of God... in beautiful things he saw Beauty itself; all things were to him good... He embraced all things with a love and enthusiasm never seen before, speaking to them about the Lord, inviting them to praise him" (2C 165; Habig 1972, p. 494). The same biographer emphasizes the mystical approach of the Saint, who "rejoiced before the flowers, and when he admired their delicate form or inhaled their sweet perfume", they transported him "immediately to the beauty of that flower which sprang from the root of Jesse in the splendid time of spring and with its perfume raised thousands from the dead... In the same way, he invited with great simplicity the wheat fields and the vineyards, the stones, the woods, and all that is beautiful in the fields, the

springs and all that is green in the gardens, the earth and the fire, the air, and the wind, so that they would have much love and praise the Lord generously. After all, he called all creatures sisters, intuiting their secrets in a special way, experienced by no one, because in truth he seemed to be already enjoying the glorious freedom of the children of God" (1C 81; Habig 1972, p. 429).

This singular ability to grasp cosmic "secrets" came to him, in the first place, from his unfiltered proximity to his *sister* creatures. That is why he can *praise* the Creator *through* and *with* Their creatures in the same act of brotherhood and gratitude. As the sources testify, the CC intends to be a correction of the habitual human ingratitude in the way of appropriating creatures, composed "in *praise of the Lord's creatures* whom we use every day, without whom we cannot live and for whom the human race greatly offends the creator", while we are "ungrateful for so many graces and benefits, not praising the Creator and giver of all goods, as is our duty" (MP 100; Habig 1972, p. 1236). In this sense, the Franciscan model of the frugal and sustainable use of natural goods can be of great interest for an integral ecology (Hubaut-Bastaire 2007, p. 110). At a time when there was still no awareness of the limits of the earth, the Poor Man of Assisi, in composing the *Canticle*, already warned of human abuses in relation to creatures "in whom the human race so offends the Creator" (LP 43; Habig 1972, p. 1022).

Such gratitude is based on the logic of the gift, which is opposed to the logic of utilitarian and mercantilist possession. The Franciscan relationship with things is neither one of domination nor instrumentalization. After eliminating all instinct of appropriation (*sine proprio*), will to power (*cum grande humilitate*, CC 14), arrogance and domination (*subditi omne creaturae*; 1R 16:6; 1LF 9: 47), and inequality (*omnes fratres*), there is only room for sympathy and cordiality with all beings. For this reason, *Laudato Si'* establishes a link between the voluntary poverty of Francis of Assisi and the critique of the technocratic paradigm (*LS*, chapter III). Since everything is a gift and there is an ontological solidarity between creatures, no one should appropriate them. "The poverty and austerity of St. Francis were not simply an external asceticism, but something more radical: a renunciation of making reality a mere object of use and domination" (*LS* 11).

*For* and *through* creatures, the troubadour of Assisi praises his Author; *with* them and *among* them, the Creator sings (LM VIII:9). Such a stance springs not only from his religious faith but also from the poetic sensibility that synchronizes him with the deepest human experience. The *Canticle of the Creatures* is an excellent expression of this, as an exponential poem of a cosmic and ecological sapiential vision that, although it springs from the faith of its author, crosses the boundaries of dogma.

There is no doubt, however, that the central theme of the CC is gratitude and thanksgiving to the divine Creator, starting from his Creatures, as underlined by the refrain repeated in each of the stanzas: *All praise be yours, my Lord.* The primary objective is to recognize and give thanks for the goodness of God in the goodness of created beings and of the entire cosmos (1R 17:17–18; 23:31–34; 2 LF 10:61–62; PBO 10; CC 14; OP 1). Many rightly see in the *Canticle of the Creatures* an anti-Gnostic manifesto aimed especially at the Manichaeism of the Cathar and Albigensian contemporaries, who saw in the universe a curse, in life a demonic manifestation, and in the beauty of the world a snare of the devil. Without naming enemies or adversaries that do not exist in his dictionary, the *Poverello* proclaims the goodness, beauty, and usefulness of a world that reflects the power, goodness, and providence of the God "most high, omnipotent, and good Lord" (Paolazzi 2019, p. 779). Correcting the long history of heretical hatred of creation, the praise to God "With all his creatures" invites us to rediscover in the sun and moon, in water and wind, in fire and earth, and in bodily death itself the light of the Creator. Moreover, it is not only the aesthetic beauty of the creatures that deserves the *Poverello*'s meticulous attention. The creatures are also sung for their ethical qualities: the "Sister Water", because she is "very useful, humble, precious and chaste" (7); and the "Brother Fire" is "beautiful, jocund, robust and strong" (8). But these two levels of meaning (aesthetic and ethical) culminate in the theological meaning of creatures: they are a mirror of the Beauty that is God: "In beautiful things he

recognized the supreme Beauty" (2C 165; Habig 1972, p. 95). Francis is the first Christian to extend the term "brethren" to inanimate beings.

Both philosophy and modern sciences are heirs of an old dualism that they insist on accentuating in splits and contrasts with these: light–dark, I–other, subject–object, body–soul, thought–matter, inner world–outer world, culture–nature, man–world, etc. From this emerges an attitude of *unconditional domination*, *conquest*, and *consumption*. Francis, on the other hand, has overcome all of these dualisms, beginning with the *sacred–profane*, proposing, in word and deed, the unique utopia of the great cosmic fraternity already announced by Isaiah (Is 11:6–10; see 1C 80–81). Unlike the Manichaeans, he not only did not allow himself or others to belittle creatures, but he also personified and dignified them, even the humblest "sensitive and insensitive" beings (1C 58; 277). Not even in the name of religion does the Franciscan universe diminish or devalue the elements of nature. These, in turn, reciprocate in the same way: "All beings strove to show their love for the Saint [Francis], to show him their gratitude: the fire, the water, the birds, the bees, the pheasant, the cicada" (2C 166–171). In the Franciscan school (St. Bonaventure), the whole of created reality constitutes an admirable ontological synthesis that expresses the benevolent presence of the Creator. God, Man, and the World, far from being rivals, are part of a harmony ordained by Being and meant to be.

*2.2. Praise Be to You, O Lord, with All Your Creatures*

We can say that the *Canticle of Brother Sun* is like the rays that spring from the same source of Beauty and Good: *the Most High, omnipotent and good Lord* with which he opens the initial doxology. The *goodness* of the Lord expands and is expressed in all creatures. *Bonum diffusivum sui*, "good [and beauty] diffuse to itself", St. Bonaventure[7]. In the *Commentary on the Sentences*, St. Bonaventure writes that "creation is like a river that gushes out of a fountain, expands through the whole earth and finally returns to its point of origin" (Delio 2003, p. 22).

*Cum tucte le tue criature*. They are all creatures *of* the Lord, who is to be praised *with* and *by* all, without exception. Love for nature, far from being self-interested or abstract, conventional, or anonymous, is marked by a respectful consideration for each being: each and every one deserves the same delicate courtesy. Therefore, after embracing *all* creatures in the same praise, he will concretize in each one. The possessive, *tue criature*, seems to us to be intentional. The creatures *are from to the Lord* and belong to Him. *Praise be to You, good Lord*, is the refrain that makes the opening verse resound*: Most High, Almighty, Good Lord.* For it is to Him, and not to creatures, that all lauds are addressed. They are the creatures who speak well of their Creator and proclaim, in their beauty, usefulness, and meaning, the goodness of the Lord (Lehmann 2011, p. 205).

From the recognition that everything belongs to the Lord, Franciscan poverty and ecology are born. To want to appropriate any creature is to want to usurp what belongs only to *bon Signor*; it is to violate the dignity of nature and of human beings in order to exploit and enslave them. This desecration, contrasted with the *sanctissime voluntati* of the *bon Signor*, Francis calls in the final stanzas *peccata mortali* (CC 13). It is "mortal" because it leads to death.

For the *Poverello*, it is fraternity that must unite creatures: *frate sole, sora luna e le stelle, frate vento, sor'aqua, frate focu, sora nostra matre terra*. In this descending order (from the sun to the earth), Francis does not see a degradation but a harmonious beauty. This universal cosmic brotherhood springs from a courteous and respectful heart towards all beings. In the Franciscan universe, all living beings are animated and personalized. Celano says that he "gave the sweet name of sisters to all the creatures of whom, by a marvelous and unknown way, he divined the secrets" (1C 81). Centuries before contemporary scientific discoveries (Crick and Dawson, who deciphered DNA), he intuited that we are related, brothers and sisters, to all beings, because we share the same basic genetic code. For him, everything is connected: "by the friendly union that he established between all things, he seemed to have returned to the state of innocence" (B 8:1). "We and all beings in the

universe, being the work of the same Father, are united by invisible bonds and form a kind of universal family, a sublime communion that impels us to a sacred, loving and humble respect" (*LS* 89). To call "brother-sister", much more than manifesting a certain tenderness and poetic sensibility, is a brilliant intuition that allowed *Poverello* to perceive a kind of blood kinship among all beings: they participate in the same material composition and the same origin in the one Creator.

In Francis's worldview, nature is not only the work of God, but it is also the great complement of Man, who is nothing without the other creatures. It is precisely from this fraternization of the world that our cosmic responsibility and mission derives. A mission that cannot be limited to technical and scientific knowledge (often conceived as a way of "dominating" nature; Russell 1982, pp. 214–15) and the consequent loss of the notion of the world as a common *home* of which humans are a part.

*Spetialmente messor lo frate sole* (3): Francis reveres, in *a special way*, the "Lord Brother Sun" and "our sister Mother Earth" *sora nostra matre terra* (9) (see MP 119), wanting to embrace all beings in the same fraternal respect that excludes ambitious domination or selfish possession. "Because he considered and said that the sun is the loveliest of God's creatures and the most worthy of comparison with God", "he had a singular predilection for the sun and fire", especially because both transform darkness into light, illuminating man, but also because they are symbols of "the Lord who enlightens our eyes through these two brothers" (MP 119; Habig 1972, pp. 1257–58). As we can see, Francis does not separate aesthetics and mysticism from usefulness and attention to reality. The sun reminds us of divinity because it renders the precious service of enlightening humanity. He himself wanted to call his poem *the Canticle of Brother Sun* because he is "the most beautiful of creatures and the one who most resembles God" (MP 119; LP 43; Habig 1972, pp. 1258, 1022) and to underline that it is the light that must be celebrated. That is why he associates the *beauty* and *splendor* of the sun with the *claritas* of the night, illuminated by the moon and the stars*: "Laudato sì, mi' Signore, per sora luna e le stelle: in celu l'ài formate clarite et pretiose et belle*" (5). According to the biblical account, in the beginning, "there was evening and morning" for every day (Gen 1:1–26). Darkness and night are not God's creatures, and, therefore, they truly are not and only exist in the absence of light. Night happens when the light goes out or we move away from it (Jn 13:30). For this reason, Francis also fixates not on night or darkness, as the Manichaeans did, but on the resplendence that, in this world, the divine Light "signifies".

The celestial bodies, the moon, and the stars are "precious and beautiful" but not divine. That is why they are also "brothers". But this does not detract from their beauty and value. We know the symbolic readings that, in the more recent tradition, associate St. Francis with the Sun and St. Clare with the moon. It is no coincidence that, as we have already recalled, this *Canticle* was born near the convent of San Damiano, where Clare and her sisters lived. It is even legitimate to suppose that they inspired the evocation of the "sister moon, clear, precious and beautiful" (5) (Leclerc 1988, pp. 95–97). As Sister Death approaches, Francis sings the *claritas* in a subtle tribute to his soulmate, Clare of Assisi, whose last words will also be a praise to life: *Praise be to You, Lord, for having created me!* (*Legend of St. Clare*, 34).

After praising the Creator *with* and *through* Celestial creatures, he praises Him *by means of* terrestrial beings, starting with the "four elements" (6–9): air (wind) (6), water (7), fire (8), and earth (9). "*Laudato si', mi' Signore, per frate vento et per aeree et nubilo et sereno et omne tempo*" (6). In the same praise, he includes "the clouds, the serene, and all the time" by whom the Creator "sustains" "all his creatures" (6). For Francis, there is no such thing as "bad weather" because, *all the time*, it is God's work for the benefit of creatures (cf. Ps 104:29–30). It is probable that, in expressing himself in this way, he is thinking of how the air/wind "brother" sustains life and the divine "breath" sustains all living beings (cf. Gen 1:2; Ezekiel 36:25–26).

Goodness and beauty are the double themes under which the *Canticle* continues, praising the Lord for *sister water*. Francis also had a special love for water, namely because it

"symbolizes holy penitence and tribulation in which the impurities of the soul are washed away" (MP 118; Habig 1972, p. 1256). That is why she deserves a generous description: "very useful and humble and precious and chaste". It is *precious* precisely because of its "usefulness". Here, too, Francis's words are prophetic. Water is, in fact, an increasingly precious and essential commodity. He calls it "humble" because it runs in the earth (humus) and accomplishes its mission and *usefulness* by always descending or trickling down from the top. This obvious detail reveals once again the depth of *Poverello*'s symbolic gaze. Finally, the water is "chaste", due to its transparent appearance and purifying effect. The water washes away and absorbs impurities. In this regard, Francis seems to warn us of the problem of the "quality of water" that only serves us to the extent that we take care of its purity or chastity, as well as of the need to "save the *precious water*". The biographer recounts that, "when he washed his hands, he was always careful to do so in an appropriate place, so as not to spill water that would be trampled by his feet" (MP 118; Habig 1972, p. 1256).

The minstrel of Assisi praises the Lord for "brother fire", which he calls "*beautiful, jocund, robust and strong*" (8). We know that he particularly admired and loved fire (MP 115; 2C 166; LM, V, 9; MP 117). He sees in him the usefulness of "lighting up the night", forgiving him the times when his "robustness and strength" are translated into violence. We know that, in medieval sensibility, fire was also associated with the demonic, connoted with hell and its punishments. The Saint of Assisi sees no hint of evil in "brother fire". As a creature, he is neither divine nor perfect, as Clement of Alexandria had reminded us long before him: "I aspire to the Lord of the winds, to the Lord of fire, to the Creator of the world, to the illuminator of the sun; I seek God, and not the works of God".[8] Francis is also a seeker of God, but he finds it in creatures, especially in his brother fire, the image of the fire of divine love (PLP 2).

*Laudato si', mi' Signore, per sora nostra matre terra*. Only to "earth" is reserved the double title of "sister" and "mother". Many cultures celebrate "mother earth" with a deep connection to the experience of mobarnabtherhood, the feminine, and the great mystery of life. Francis also knows that we are earth and, at the same time, children of the earth, in a double sense: as Adamics, but also insofar as it is from the earth that everything comes to us—"*la quale ne sustenta et governa, et produce diversi fructi con coloriti flori et herba*". Like a good mother, the earth *nourishes* her children with reliable generosity.

Francis adds that the earth not only *nourishes* us but also *governs* us: *La quale ne sustenta et governa* (9). Again, the perspective is reversed: it is not man who "governs" the earth, but the earth "who governs us". The laws of this common mother and home must therefore prevail over human and political laws. Therefore, before we take care of the earth, it is the earth that takes care of us. Taking care of the earth is nothing but giving back for what this "mother" does for us. In addition to "feeding" us with "diverse fruits", the earth, in its maternal facet, exceeds itself by offering us the beauty of its "colorful flowers and greens" that are the ornament of this sister and mother. Once again, nature, far from being a reservoir of resources to be enjoyed, is a garden to be contemplated and cared for, without ceasing to be useful, by "nourishing" us. We are alert for the link between "government", "sustenance", and sustainability. If this last concept is not yet part of our concerns, there is no doubt that, with "mother earth" being the subject of the verb "to govern", sustainability is implicit.

The common earth that we inhabit is like a *sister* with whom and in which we share existence, but it is also a beautified mother (cosmos) who treats us well. Pope Francis made this eighth stanza of the *Canticle* the great motto of his integral ecology, recalling that "in this beautiful song, Francis reminds us that our common home is also like a sister with whom we share our existence; and like a beautiful mother who welcomes us into her arms" (*LS*, 1). In *Evangelii Gaudium*, the Pope had already adopted the language of Francis of Assisi to recall that "the earth is our common home and we are all brothers and sisters" (183). Instead of condemning the radical ecology that has restored the sacralization of the ancient goddess *magna mater* (Gaia), the Bishop of Rome prefers to positively take up the



Christian proposal of St. Francis. By calling her both "sister and mother", the author of the *Canticle of the Creatures* sets a limit to the "motherhood" of the earth. She is our "sister" because, as the Judeo-Christian Scriptures teach, man "was taken from the earth and to the earth he will return" (Gen 2:7; 3:19). In fact, we too are "earth".[9]

We underline the strong association of the designation of the earth as "sister" and "mother" with the notion of "common home". According to biblical tradition, God "created the earth, not to be desolate, but to be inhabited" (Is 45:18). In this common home, we can only live as brothers and sisters, children of the same Father and the same Mother, who nourishes us with her fruits. What the ancient Romans called *oikouméne* continues to serve as a fundamental concept for an ecological, religious, and para-religious ecumenism of which Francis could also be the patron (cf. Armstrong 2023, p. 132).

Thus, the Saint of Assisi goes far beyond the cosmic optimism of the ancient Greeks, recovering the luminous meaning of creation (Gen 1:31; cf. St Bonaventure, *Itinerarium*). With the cosmos, it is possible to re-establish a fraternal and "maternal" relationship. The world ceases to be evil, adverse, or an accumulation of dark and blind forces and becomes an open book in which we see ourselves as brothers and sisters, children of the same Father and of the same Creation. Material and inanimate things become signifiers of the "sign", "image", and "presence" of Someone related to us. The planet ceases to be a reserve to be exploited, to become again the garden with useful and useless, but always beautiful, things: *con coloriti flori et herba.* From this, our saint deduces that the world has been entrusted to us not only to "use" it but also to admire and care for, just as the gardener cares for his plants. For this reason, St. Francis asked his brother, who took care of the garden, to always leave a portion of land uncultivated, so that weeds and flowers could also grow freely (2C 165; MP 118). This shows that, in his mind, two great misunderstandings have been overcome: the Manichaeism that divides the world into "good" and "evil"; the economic reductionism kills or prevents the different and the smallest from growing.

On the other hand, in the CC's view, there is no room for pantheism: the world and matter are never evil, nor can they be seen as demonic, but neither can they be understood and treated as divine, since they did not *necessarily spring* from God, as if the cosmos was part of the divine world, but were *created* by the one who is *the Supreme* and *Most High Good*. God is not the immanent principle or soul of a necessary world, nor its intrinsic rationality (*anima mundi*). In the ecology of Francis of Assisi, the cosmos never becomes God, and God never becomes the world. They are profoundly related realities, precisely because they are distinct: it is this difference that allows creatures to be "symbols" or "vestiges" of the Creator; they are just as distinct is man, though he is a brother to all creatures. The singer of creatures does not get lost in the cosmos or become confused with creatures. By calling "brother-sister", it shows closeness and distinction at the same time. By contemplating the "Most High" God in his creatures, Man finds his place, re-dimensioning himself in the confrontation with the Creator and with the other beings of the cosmos. Francis never confused God or himself with nature. Such confusion is as strange to him as the alternative or opposition of the dualists. The nostalgic romanticism of an "integrated" nature or the romantic inebriety of fusion with the cosmos has nothing to do with Franciscan intuition (Gonçalves 2014, p. 474). Francis's fraternization with nature is the result of an interior detachment and openness to God. It is not the wind, the fire, and the water that are the ultimate object of his praise; it is the Creator of all.

## 3. Epilogue: The Final Reconciliation

Francis saw fit to add to his *Canticle of the Creatures* the praise *for those who forgive* (10). It is important to remember that this last stanza was added a few weeks later to reconcile the bishop of Assisi with the political authority (*podestà*) of this city. It is, therefore, an exhortation in the form of praise to the promoters of peace and forgiveness, whose scope goes far beyond circumstances. Francis's entire life and proposal focused on the reconstruction of human *fraternitas* rooted in the ideal of peace, without ignoring its tensions and wounds. By integrating into his *Canticle* all the dimensions of the created world—stars,

meteorological phenomena, the elements (earth, water, fire, and air), the earth with its fruits and vegetation, human beings, and death itself—the universal Brother does not ignore dissonances. However, they do not prevent him from believing in the political implications of such a proposal, as his *Letters: to all those who live in the world beyond and to the rulers of the nations* attest, motivated by the hope of involving everyone in this great project. It took 8 centuries to see a Pope adopt a similar attitude and methodology.

Francis's approach is "poetic" in the Greek sense of *poiêin*. As such, the *Canticle* takes us back to a political dimension that leads us to policies which are not just focused on human and social development but are projected towards a planetary vision that is concerned about future generations and encompasses all the nonhuman realities of the planet. St. Francis's radical poverty is nothing more than an alternative way of respecting all beings and the planet itself as a heritage that is neither our "domain" nor the heritage of a people or a generation (cf. LF 33; Habig 1972, p. 1328).

If the entire CC is a pact of reconciliation with all creatures, humans could not be left out. In Francis's universe, there are, as we have said, no dualisms or dichotomies between man and nature; humanism and naturalism; and environment and community. He knows that "humanity is not the only expression of being", but he also knows that it occupies a unique situation in the context of Creation (Gonçalves 2014, p. 297). That is why human beings also enter into *the praises* of the *Canticle of the Creatures*, but always in correlation with all other beings. Man is a cause for praise to the extent that he *forgives*: *Laudato si', mi' Signore, per quelli ke perdonano per il tuo amore...* (11); and they are *peaceful people* who "endure tribulations" and "sufferings" (11). The cosmic and universal fraternization proposed appeals especially to the social and ethical responsibility of man in his relationship with others. As such, it requires a new attitude that involves the whole of human life, cosmic, social, and moral. Francis advocates for us to overcome the conflictual anthropological and political model *(homo homini lupus)* that is at the basis of modern culture (T. Hobbes), which does so much damage to society and to our common home, and to show us the way to reconciliation and peace, which requires the exercise of patience: *...et sostengo infirmitate et tribulatione* (10). Therefore, it teaches us that, without forgiveness, there is no fraternity or common home, and without community, there is no ecology. The proposal that comes to us from Assisi is also a pedagogy for a healthy coexistence of the human family. The environmental crisis that emanates greatly from an arrogant anthropocentrism and the consequent anthropomorphization of social organization will only be overcome by acknowledging the errors and offenses committed by humans.

The peace pact proposed is born of the renewed man and embraces all dimensions of life, including death, which ceases to be the "monster" and taboo and returns to being the twin "sister" of life (Gonçalves 2014, 308ff.). For this reason, the CC ends with praise "for sister bodily death": *Laudato si, mi Signore, per sora nostra morte corporale* (12).

Thomas of Celano reminds us how Francis "spent the last few days of his life in a hymn of praise, inviting his companions to the same praise... He also invited all creatures to praise God... Finally, to praise his own death, terrible and hateful for all, and to meet it joyfully, inviting her to be his guest: 'Welcome,' he said, 'my sister Death'" (2C 217; LP 65; Habig 1972, pp. 536, 1042). He referred to her as "sister" because she relates us to all finite beings.

We better understand the meaning of the text if we read it in the light of the last gesture of Francis, who, already dying, asked to be stripped and placed naked on the ground, in physical unity with the *sister earth* (2C 217; Habig 1972, p. 538). He wanted to die in the arms of *Mother* Earth, assuming fully his finite identity with her. Francis communicates once again with God through the earth, also reuniting in death with all beings: "*Hebraicum Adam in latino interpretatur terra caro facta*", writes the anonymous author (IIIth century), *De montibus Sina et Sion* (4).

The hagiographers also add that, as he sensed "sister death" approaching, Francis wished to see his Roman friend Jacoba of Settesoli again and to taste, for the last time, her delicious Roman biscuits (*mostaccioli*) (MP 112). The originality of the Saint surprises us

once again. Like a pregnant woman who, when she dies (true childbirth), has less than holy desires, the Poor Man of Assisi wants to say goodbye to the earth by savoring a delicacy that reminds him of how sweet friendship is—a final proof of the tenderness and humanity of someone who, far from wanting to make a gaudy or even miraculous spectacle out of death, wanted to teach us how to die well, after having taught us how to live. After all, what is more "common" and familiar than death? Francis recovered his life, but he also recovered death, uniting it with life. For him, death is the summit and door of life, and all life is a path of uninterrupted encounters, the last of which with the most inevitable sister.

As the philosopher Joaquim Gonçalves (2014, pp. 304–15) shows us, it will be very difficult to have an ecology of life without an ecology of death. Here, the lesson of Assisi is pertinent and timely too. Only when we move from an ecology based on the fear of dying (a matter of survival) to one grounded in living like "mortals", will we be able to practice integral ecology. In fact, those who understand life as a *praeparatio mortis* and death as an Easter/passage will be much more environmentally friendly and "brotherly" to all.

If Francis's final *Canticle* has the morning splendor, it is precisely because it appears in the evening of a life profoundly lived. As such, it can be the *song* of all humans, which must end, after all violence, with the final stanza of all pardons. As the poet writes, Francis's *canticle* "celebrates the inaugural/Beyond the death of the gap of loss and disaster/Your poem salutes the first truth of every creature/The wholeness of the initial day" (Sophia de Mello Breyner, *Poema: São Francisco*).

The CC closes with a call to lauds and *gratitude*, to which is added the call to "serve [the Lord] with great humility": *Laudate e benedicete mi' Signore et ringratiate e serviateli grande humilitate* (14). It will not be forcing the meaning of this concluding stanza to see in it a call to care for creatures and for the common home, understood as "service to the Creator". Franciscan ecology does not set itself apart by striving for the preservation of species but by "service" to all creatures, respecting their difference and intrinsic value. This is incompatible with the exclusive protagonism of humans as the center and owners of the world.

Western culture has long been accustomed to looking at nature as inhospitable and tamable through technology, which, in turn, has imposed a mechanical view of the universe, as if it were separate and independent of those who inhabit it. Francis understands the world as a dwelling place, and because to understand is already to build, he is undoubtedly one of the builders of the common home (Merino 1991, pp. 176–81) by establishing a new relationship with creation (see 1C 81; MP 115; 118). God, creation, and man are neither independent nor adversaries but realities in fraternal relationship, without being homologous or confused. Thus, Francis's grammar goes beyond not only the paradigm of *domination* but even the *administrative model* in the reading of Genesis and of man's relations with the created world to propose the model of *kinship*.

If Francis has been proclaimed the patron saint of ecologists (and not of "ecology"), he can continue to inspire those who dedicate themselves to the protection of our common home. To this end, it would be important to study and learn more about the Franciscan tradition and others that contradict the *Communis Opinio*, which makes the Christian tradition the great cause of the ecological crisis. In this context, the CC can be considered a very valuable manifesto or program for an integral and inclusive ecology (*LS* 93). To this end, it will have to be read and interpreted in light of the symbolic gestures of the Saint of Assisi, as we have tried to do. In these gestures, we can learn a new vision on the world of perennial value, regardless of the creeds professed or refused. In fact, the Franciscan intuition is inscribed, in its radicality, in the tension between ethics and aesthetics that explains its timeliness (see Poitrenaud-Lamasi 2020, p. 229).

**Funding:** This research received no external funding.

**Institutional Review Board Statement:** Not applicable.

**Informed Consent Statement:** Not applicable.

**Data Availability Statement:** Data are contained within the article.

**Conflicts of Interest:** The authors declare no conflicts of interest.

### Acronyms

| | |
|---|---|
| 1C Thomas of Celano | *First Life* |
| 1LF Francis | *First Letter to the Faithful* |
| 1R Francis | *First Rule* (12021) |
| 2C Thomas of Celano | *Second Life* |
| 2R Francis | *Second Rule* (1223) |
| CC Francis | *Canticle of the Creatures* (*Canticle of Brother Sun*) |
| Ex Francis | *Exhortations* |
| LF | *Litle Flowers* |
| LM St. Bonaventure | *Major Legend* |
| Lm St. Bonaventure | *Minor Legend* |
| LP | *Perusina Legend* |
| MP | *Mirror of Perfection* |
| OP | *The Office of the Passion* |
| PFO Francis | *The Praises before the Office* |
| PG Francis | *Praises of God* |
| PLP Francis | *Paraphrase of the Lord's Prayer* |
| T Francis | *Testament* |
| TC | *Legend of the Three Companions* |

## Notes

[1] It began with John XXIII, but especially with Pope John Paul II, who proclaimed the *Poverello* Patron of ecologists. Cf. John Paul II, *Apostolic Letter Inter Santos*, 29 November 1979.

[2] Jacques of Vitry, *History of the Orient*, 32; (Habig 1972, p. 1610).

[3] This Byzantine-style image or icon, perhaps the most reproduced crucifix in our day, is preserved in the Basilica of St. Clare of Assisi, where it was transferred when the Poor Clares left San Damiano (Accrocca 2007, pp. 407–50).

[4] José Saramago, *A segunda vida de São Francisco*, (Saramago 1987, p. 51) ("When, in order to surrender myself to poverty, I renounced my inheritance, I also renounced a father who had nothing more to give me than these goods of vanity. For him, the inheritance and the son were the same. Because the son did not want the inheritance, the inheritance no longer wanted the son. Listen to the word of money...").

[5] Cf. J.P. Vivet, *Les mémoires de l'Éurope*, I, (Leclerc 1970, p. 353; Le Goff 2007, pp. 154–55).

[6] This is how Pope Francis himself called it in his *Message to Muslims throughout the World for the end of Ramadan ('id al-fitr)*, 10 July 2013.

[7] Bonaventure, *Comm. Lc.*, XI, 34 (7, 287); Thomas Aquinas, *Summa Theologica*, I, p. 1, inq. 2, tract. un., q. 1, tit. 1, c. 1, arg. 2.

[8] Clement of Alexandria, *Protrepticus*, VI, 67, 2 (Claude Mondésert, Clément d'Alexandrie, Le Protreptique, Paris: Du Cerf, 2004), p. 132.

[9] See *Pseudo-Barnabas*, VI, 9 (P. Pringent-A. Kraft, Épître de Barnabé, Paris: Du Cerf, 1971), 122; John Chrysostom, *Catechesis*, II, 5: "you are earth" (Antoine Wenger, *Jean Chrysostome. Huit Catéchèses Baptismales*, Paris: Du Cerf, 2005), p. 136.

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
