# Peer review of "The Canticle of the Creatures by Francis of Assisi (1181/82–1226) and the Care of Our Common Home"

_religions, doi:10.3390/rel15020184_

Round 1
Reviewer 1 Report
Comments and Suggestions for Authors
Very good article, profound, convincing, documented.
Perhaps the Author should have provided a less reassuring notion of aesthetics in the paragraphs “Aesthetics and ecology”, since in the contemporary context the patterns of what constitutes beauty is highly contested, starting from the famous statement “The beauty will save the world” that Dostoevskij attributes to prince Miškin (novel “The Idiot”). In fact, it is more a question that a statement. In Christianity, the beauty has a direct relation with the Good, and for that reason beauty doesn’t coincide with the meaning that it takes in ancient classical cultures. Beauty is not symmetry, for instance, and it relates more to ethics than aesthetics.
A second comment regards the “joyful” approach to life of Francis, which is certainly true, but it doesn’t come without a cost. From this angle, we might discover that beauty can also be ugly, implying pain and suffering. In fact, the other side of Franciscan aesthetics is illustrated by his Stigmata. In that sense, we could also infer that ugliness will save the world.
In sum, I would have loved to read more about a tension, a polarity, in the world vision of Francis, beyond the celebration of his inclusive concept of creation. Morevover, beyond contemplation, Francis’ concept is “poetic”, in the original Greek etymology of the word, implying “poiêin”, doing real things, dealing with the material world. The Canticle should also suggest a political approach: world politics is not enough from the point of view of the integral ecology, since what we really need is a planetary politics, capable of including non-human components of the planet, such as plants and animal, as well as future generations.
Author Response
I tried to intervene in the text in order to respond to the reviewer's constructive criticism. As you'll see in the new version; I've even revised the entire Pinglish translation/version.
Reviewer 2 Report
Comments and Suggestions for Authors
The Canticle of the Creatures by Francis of Assisi and the care of our Common Home
Interesting article that establishes, in a very justified way, the direct relationship between the proposal of closeness and brotherhood between the Hymn of the Creatures of St. Francis and the proposal of integral ecology of Pope Francis, especially in his Laudato si. With a very exhaustive argumentation, both the life of the Saint of Assisi and his naturalist message are explored in depth. The work presented demonstrates this connection between the two Francis, which the Jesuit pope himself recognises perfectly.
It is also interesting, in the first part of the article, to note Francis' own vocation with his memorable "rebuild my house", which refers both to God's house and to the house of all. This vocation serves as justification for his Hymn to Creation and its creatures.
Based on this assessment, I consider it necessary for the author(s) to reflect on the following points in order to give more depth to the work presented:
1. Necessary modification. A better explanation of the objective of the work, which seems to be the relationship between the two messages, but a clear definition of the objective would be necessary in order to be able to check to what extent it is achieved.
2. Necessary modification. All references should be revised, both in the text and in the references section. Although the work needed is exhaustive, as a summary, the following should be modified:
a) Textual citations. Where the author or authors, year and page should appear.
b) Indirect quotations. Where at least the author or authors and the year should appear.
c) The formulation of the references must also follow the APA 7 model, where, for example, it is not necessary to include the cities of the books or texts.
d) It would be advisable for the author(s) to review these rules and to consult some articles in this journal.
3. Proposal for improvement. The texts included in other languages (Latin) are very interesting, but they would need to be translated into the language of the article to make them easier to understand.
4. Proposal for reflection. The parallelism between the Hymn to the Creatures and the encyclical letter Laudato si is a little lacking when we talk about the proposal of Pope Francis in his integral ecology, which also includes the human being. As another creature, but with political, social and religious responsibility. The human being must also be a creature who deserves respect and who occupies a special place within the whole of creation because of his creation in the image and likeness of the Creator.
I would like to congratulate the author(s) for the work presented for the completeness of the article and for the close relationship between the Saint's vocation, his proposal towards Creation and the Pope's proposal of integral ecology.
Author Response

(The authors gave the same response as above.)

Reviewer 3 Report
Comments and Suggestions for Authors
Most important critique is that the question is not clearly stated, not developed and therefore not sufficiently answered. I think the question is about the relevance of Franciscan spirituality, especially the Canticle of the Creatures, and perhaps also Franciscan aesthetics, for late modern ecology. And as I understand it, the author wants to move from "repair My House" via different extensions of "brotherhood" until the "boundless brotherhood of the good Creation" to what it really means: "My House," especially in ecology of today. Many remarks of the author are very fine - he or she really understands the Canticle! - , e.g. that sister mother Earth governs us instead of the Biblical image of Adam governing the earth (although the Umbrian governa is not the same as "subdue and rule" in Genesis 1:28), that the praises are given through and by the creatures, that Francis does not own or govern anything in the world, that ecology matches with aesthetics and penitence, that we are in a universal brotherhood and goodness. Very good, I agree, but the point should be made in an argument, not a testimony.
My second critique follows from this. For although I agree with the content of the article - "essay" - it is not written as a scientific investigation. It is a testimony, uses too many words to stress its points, but seldomly investigates the original text. For example, it says that "he can praise the Creator through and with Their creatures" (l. 333) but does not discuss the grammatical phrase "Laudato si' ... per ..." nor does it refer to an existing analysis of the text. It says that Francis was "removed from the leadership of the Order" (l. 254) without discussing his own withdrawal in 2C 143. It says "without ignoring the 'Franciscan question'" but never discusses the value of a testimony even from the official biographer Thomas of Celano, who is also an interpreter. It says "As well illustrated by the Florinhas (Little Flowers), a poetic and symbolic interpretation of the Franciscan persona, the sensitive soul of the Poverello ...," (l. 317-318) and "we can see in this Canticle the music score that every human being is called to rehearse in treble clef" (l. 227) and later "inspired by and in G major" (l. 261) but this overly exuberant language does not match a scientific article.
Furthermore, the formulations are not very accurate. If you quote Paul Ricoeur, who, by the way, is quoted by Eloi Leclerc, you should write your reference in a footnote (and please, explain what the authors want to highlight) (l. 233). The critique on modern sciences is not untrue (ll. 379-382), but to write it in three lines without reference is way too inaccurate. Thomas Aquinas would not support "bonus diffusivum sui" as a one-liner (see Dewan, L. (1978). St. Thomas and the Causality of God’s Goodness. Laval théologique et philosophique, 34(3), 291–304. https://doi.org/10.7202/705686ar) and the Latin is not translated as "good [and beauty] diffuse to itself" (l. 399). Beauty is already a name for God (l. 201) in Augustine's Confessions (X,38) and Augustine shows that this Beauty is not to be confused with all forms of beauty before our eyes and ears, as does Pseudo-Dionysius in Mystical Theology. Thomas of Celano did not write "he called all creatures sisters" (l. 329) as 1C 81 says "Omnes denique creaturas fraterno nomine nuncupabat". Paolazzi did not publish in 2029 (footnote 10).
My suggestion: 1) remove as much as possible poetic and enthusiastic language and 2) develop the line of your thought through Francis' personal development regarding the growth of his brotherhood and indeed reparation of Gods House, by the way ending in Greccio's manger; 3) give all the references needed (where is it stated that Thomas is "the most faithful biographer" or that 3Companions is a "most reliable source"?), 4) discuss matters critically (is repairing churches only "playing the part of a bricklayer" or was it also a penitence for converted cathars and people who failed to engage in a crusade e.g. in Apulia? Was Francis really aware of the depth of his mission, as Chesterton writes?) 5) and follow your reading - which is good - of the Canticle itself closely. And finally, what is it that late modern ecologists can learn from the Canticle? Would it be the quotation from LS 12 in line 64: that ecologists may stop being stuck in problems-to-be-solved and start to open themselves for the mystery, namely that the Earth can manage all problems herself as she has done for the past several billion years?
Comments on the Quality of English LanguagePerhaps an artificial translator has been used, but AI is not always trustworthy. For example, what does "gathering the stones that some of his Give, other you Shoot" (l. 112) mean? Or: "Many rightly see in the Central Committee an anti-Gnostic manifesto" (l. 364)? Is "jocund" (l. 374) an English word? What is "iuvare, iucundus, usually said the Latins" (l. 308) mean? "Communis Opinion" (l. 659)?
Author Response

(The authors gave the same response as above.)

Round 2
Reviewer 3 Report
Comments and Suggestions for Authors
Much better. I still have some questions, but I am always grateful when an article raises questions. Thank you.